# The Use of Psychotropic Medication in Pediatric Oncology for Acute Psychological and Psychiatric Problems: Balancing Risks and Benefits

**DOI:** 10.3390/children9121878

**Published:** 2022-11-30

**Authors:** Johanna M. C. Blom, Elena Barisone, Marina Bertolotti, Daniela Caprino, Monica Cellini, Carlo Alfredo Clerici, Chiara Colliva, Cinzia Favara-Scacco, Silvia Di Giuseppe, Momcilo Jankovic, Alessia Pancaldi, Luca Pani, Geraldina Poggi, Veronica Rivi, Fabio Tascedda, Riccardo Torta, Dorella Scarponi

**Affiliations:** 1Department of Biomedical, Metabolic and Neural Sciences, School of Medicine, University of Modena, 41125 Modena, Italy; 2Center for Neuroscience and Neurotechnology, University of Modena and Reggio Emilia, 41125 Modena, Italy; 3Pediatric Hematology and Oncology, Azienda Ospedaliero-Universitaria, Città Della Salute–OIRM, 102126 Turin, Italy; 4Pediatric Hematology and Oncology, Institute Giannina Gaslini, 16147 Genoa, Italy; 5Department of Pediatric Oncology, Azienda Ospedaliero-Universitaria Policlinco di Modena, 41125 Modena, Italy; 6Department of Hematology and Oncology, University of Milan, 20126 Milan, Italy; 7Pediatric Oncology Unit, Fondazione IRCCS Istituto Nazionale dei Tumori, 20133 Milano, Italy; 8Local Health Unit of Modena, District of Carpi, 41012 Carpi, Italy; 9A.O: U. Policlinico-San Marco Psychotherapy & LAD ONLUS, 95123 Catania, Italy; 10Center for Pediatric Hemato-Oncology, Marche Polytechnic University, 61100 Ancona, Italy; 11Paeditric Clinic, Universityof Milano-Bicocca, Fondazione MBBM, 20900 Monza, Italy; 12Department of Psychiatry and Behavioral Sciences, University of Miami, Miami, FL 33136, USA; 13Scientific Institute, IRCCS E. Medea, Bosisio Parini, 23842 Lecco, Italy; 14Department of Life Science, University of Modena and Reggio Emilia, 41125 Modena, Italy; 15Department of Neuroscience, University of Turin, 10124 Turin, Italy; 16Unità Operativa Pediatria Pession, IRSSC S. Orsola, 40138 Bologna, Italy

**Keywords:** pediatric oncology, psychotropic drugs, antidepressant, mental health

## Abstract

Severe acute behavioral and emotional problems represent one of the most serious treatment-related adverse effects for children and adolescents who have cancer. The critical and severe nature of these symptoms often makes necessary the use of psychotropic drugs. A working group composed of experts in multiple disciplines had the task of creating an agreement regarding a management plan for severe acute behavioral and emotional problems (SABEPs) in children and adolescents treated for cancer. To obtain global information on the use of psychotropic drugs in pediatric oncology, the working group first developed and mailed a 15-item questionnaire to many Italian pediatric oncology centers. Overall, an evident lack of knowledge and education regarding the use of psychotropic medications for the treatment of SABEPs was found. Thus, by referring to an adapted version of the Delphi method of consensus and standard methods for the elaboration of clinical questions (PICOs), the working group elaborated evidence-based recommendations for psychotropic drugs in the pediatric oncology setting. Furthermore, based on a thorough multivariate analysis of needs and difficulties, a comprehensive management flow was developed to optimize therapeutic interventions, which allows more accurate and efficient matching of the acute needs of patients while guiding treatment options.

## 1. Introduction

Each year, in Italy, approximately 1400 children (0–14 years of age) and 800 adolescents (aged 15–19 years) are diagnosed with cancer [1,2]. Regardless of the enormous progress in treatment and enhanced overall survival, children and adolescents often face a high burden of disease, which is related both to the type of tumor, as well as the related treatment regimen, which frequently influence their physical, mental, social, behavioral, and emotional health [3,4]. Consequently, controlling efficient bio-psycho-social symptoms is the foundation of an elevated level of oncological care.

Among the clinically most serious treatment-related adverse effects for children and adolescents who have cancer are severe acute behavioral and emotional problems (SABEPs), including delirium, anxiety, and major depressive disorders [5,6,7]. SABEPs can be highly distressing not only for the (young) patients but also for their caregivers and the medical staff, representing a global health priority [5,6,7].

Furthermore, insufficient recognition or lack of adequate treatment of the symptoms of these disorders may cause or exacerbate possible complications, interfere with efficient treatment, increase the number of days spent in the hospital, and result in adverse effects and compromise short- and long-term overall outcomes [8,9,10]. Not least, these problems possibly interfere with the oncological treatment and may put children and adolescents at serious risk [11,12,13].

Thus, the early recognition and identification of acute behavioral toxicity might provide the pediatric oncologist with a timely chance to intervene (i.e., adjusting or changing the therapeutic regimen) and prevent further worsening of the extreme distress of the patients. The reader is directed to Appendix B (Figure A1) for a clinical presentation of SABEPs in a pediatric oncology setting.

While non-pharmacological interventions would constitute the preferred approach, the acute and severe nature often makes the use of psychotropic drugs necessary [14,15].

However, the lack of guidance, especially concerning the use of these medications, represents a significant issue for successful interventions. To date, no randomized trials, guidelines, or expert consensus exist in the pediatric oncology setting regarding the use of psychotropic treatment in managing severe acute behavioral and emotional problems [16]. Furthermore, the existing literature consists of studies with small sample sizes or case series and mainly considers the administration of psychotropic drugs in a non-acute setting [16,17,18].

Given the lack of evidence-based studies and consensus guidelines that assist the pediatric oncologist, as a rule, they use the medications with which they are most familiar. However, these may not offer the best option or even refrain from treatment.

In this complex scenario, an interdisciplinary working group of experts was formed with the overall aim of developing consensus recommendations regarding the use of psychotropic medication for SABEPs in the pediatric oncology setting and suggest solutions for the main problems encountered by pediatric oncologists. This paper describes and summarizes the results of the different rounds of consultation of the working group, which led to the development of a comprehensive management plan for the treatment of SABEPs and evidence-based recommendations for psychotropic drugs in the pediatric oncology setting.

## 2. Methods

The study started because of the many problems encountered in a clinical setting and reported in a non-formal way. As a consequence, a working group on the use of psychotropic drugs in pediatric oncology was formed to present a comprehensive management plan for the treatment of SABEPs, with a specific focus on the use of psychotropic medications for instantaneous and short-term therapies. The premise of the working group’s efforts was that the early recognition of patient-specific risk could help choose the most suitable patient-specific treatment for pediatric cancer patients [19,20].

### 2.1. Working Group and Study Design

An adapted version of the Delphi method of consensus was used. An intersociety working group (Italian Society for Psycho-oncology—SIPO and the Italian Association for Hematology and Pediatric Oncology—AIEOP) was created and it was enriched with experts from multiple disciplines. Experts included specialists in pediatric oncology, pediatric and adult psycho-oncology, neurology, pediatric neuroscience, psychiatry, (neuro)psychology, and neuropharmacology and were selected based on their expertise and geographic representation. The working group had the task of creating a consensus regarding a management plan for SABEPs in children and adolescents treated for cancer [14]. This group met in a first round of consultation and decided that a questionnaire was needed to better understand the entity of the problem.

Thus, a 15-item questionnaire (adapted from the questionnaire used by Kersun and Kazak (2006)) was developed and sent out to 16 pediatric oncology centers. The participating centers covered the entire national territory (i.e., Emilia-Romagna, Lombardia, Marche, Piemonte, Liguria, Campania, and Sicilia) and reflected typical regional variability in center size and patient characteristics. Pediatric oncologists and child mental health specialists were asked to complete the questionnaire enquiring about their opinions and current practice of prescribing psychotropic medications for the treatment of severe acute behavioral and emotional problems, such as depression, anxiety, delirium, and psychosis, in children and adolescents suffering from cancer [21]. Details about the questionnaire and outcomes are included in the Appendix A.

After careful analysis of the literature and the existing guidelines, as well as direct input from the questionnaire, a second round of consultation was performed, and clinical questions—reported and discussed below—emerged. Thus, the experts in the working group were asked to answer the clinical questions, providing detailed and updated information. A final round of consultation was then performed to develop the recommendations reported in this study and to bring attention to possible barriers and problems related to this complex topic.

### 2.2. Development of Relevant Clinical Questions to Guide Evidence-Based Recommendations

The working group used standard methods for the elaboration of clinical questions (PICO, PICOS [22]), which then led to the development of evidence-based recommendations for the use of psychotropic drugs in the pediatric oncology setting [23,24,25]. The target population for which the recommendations were developed includes children (aged 0–14), adolescents (aged 15–19), and, occasionally, young adults exposed to a pediatric treatment regimen.

### 2.3. State of the Art

A review of the literature was performed to provide an overview of the current state of the art, including the presence of barriers and problems.

## 3. Clinical Questions

In the pediatric oncology setting, delirium, psychosis, depression, and anxiety commonly require consultation and psychopharmacological medication [26,27]. Perhaps, in an ideal world, pediatric psychopharmacology would have a range of treatments to offer for these problems, based on extensive data, indicating the safest and most effective drugs while counting on a solid consensus on what to use, when, and for whom.

However, we do not live in such an ideal world. Instead, we must deal with prevalent, often extremely acute, or sometimes chronic mental health problems in these young patients, causing substantial functional impairment. As emerged from discussions within the working group and confirmed by our questionnaires, this scenario is further complicated by the absence of reliable data and therapeutic knowledge. Treating SABEPs in our less-than-perfect world, what to use, when, and for whom should be based first on urgency and the importance of symptoms.

Currently, there are no clear indications approved by the Food and Drug Administration (FDA) [28] or European Medicines Agency (EMA) [29] for psychopharmacological medications in medically ill children with delirium, depression, or anxiety [30,31,32,33].

Furthermore, a small group of patients may come with preexisting behavioral problems, such as hyperactivity attention disorders, which are treated with psychostimulants and other compounds [34,35,36]. As for the other psychotropic medications, pediatric oncologists need to pay attention when selecting the appropriate therapy, especially in relation to possible interactions with the cancer treatment proposed, avoiding that the combination of these treatments creates SABEPs [37].

Thus, after careful analysis of the literature and the existing guidelines, as well as direct input from the questionnaire, we formulated several clinical questions that may help to inform policy and recommendations. One of the indications we used was the needed number to treat (NNT), which in the pediatric setting and, more importantly, in specific clinical realities, such as that of pediatric oncology, has an added value because it gives a clear idea of the benefit of treatment to the individual patients [38].

The NNT provided a way to assess the impact of therapy by estimating the number of patients that need to be treated to have an effect on one patient. Especially in this vulnerable patient group, this concept might be helpful, as we know that not everyone is helped by just medicine but might need concomitant psychotherapeutic interventions as well [19,39]. In other words, some children might benefit, while some might have no benefit, but we must avoid that some are harmed at all costs. In this acute setting, an NNT of two or three indicates that treatment should be considered quite effective [40].

The formulation of the following questions—to which we provide a conditional reply—was guided by data from the literature and the results of our questionnaire.

Furthermore, to assist pediatric oncologists and other healthcare professionals in identifying symptoms of delirium, mood disorders, and anxiety disorders in this patient population, we created specific clinical management workflows (Appendix C, Appendix D, Appendix E, Appendix F).


*Q1: What is the risk of suffering from cancer-related severe and acute behavioral and emotional problems in children and adolescents: is the treatment phase critical?*


Some evidence suggests that the early phases of treatment are associated with an increased risk for SABEPs, including chemotherapy, radiotherapy, surgery, and bone or stem cell transplantation [41]. Furthermore, there is evidence that children and adolescents treated with corticosteroids have an increased risk for SABEPs [42,43,44,45,46]. Moreover, children and adolescents treated with higher doses of cranial radiation showed an enhanced risk of developing SABEPs [47,48]. Thus, to assist pediatric oncologists and other healthcare professionals in defining the psychopathological risk of children and adolescents in pediatric oncology and the choice of psychotropic drugs, a specific risk-based management flow was created (Appendix F).


*Q2: What is the influence of age at treatment on the risk of SABEPs in children and adolescents with cancer?*


Evidence suggests that patients younger than six years or older than 11 years are more at risk for SABEPs [49,50,51]. Adolescents and young adults (up to 21-years-old) treated with pediatric protocols for hematologic malignancies were particularly vulnerable to suffering from psychological problems [52,53,54]. In particular, more than one-third of these patients experience depression or anxiety [55]. Furthermore, anxiety was two times as common for patients in treatment when compared to early survivors [56,57].


*Q3: What should be done when severe acute symptoms are found?*


First, a consensus exists on the necessity to screen children and adolescents consistently for symptoms of distress based on the severity and the acute nature of behavioral and emotional symptoms; the patient should then be directed to the most appropriate treatment. See Appendix B and Appendix E for symptom-specific decision flowcharts.


*Q4: What is the best available research evidence on preventing and treating delirium in pediatric oncology patients?*


The pharmacological approach to prevent and/or treat delirium may be driven by different goals:

(1) prevention through controlling risk factors.

(2) management of delirium-related symptoms (i.e., psychosis or agitation).

(3) resolution of the underlying cause of delirium or modulation of the neurochemical cascade [58,59]. However, to date, a clear strategy to prevent and manage delirium in oncopediatric patients has yet to be adequately studied.

In general, the treatment of delirium requires an accurate diagnosis, continued monitoring and re-evaluation, therapeutic intervention for hyperactive manifestations that may cause the patient damage, avoidance of risk factors that may worsen delirium severity, and, if possible, diagnosis and treatment of the underlying etiology [24,26,60,61,62,63,64] (Table 1). Haloperidol is the most frequently prescribed antipsychotic for the treatment of delirium [65,66,67,68,69]. The successful treatment of hyperactive delirium in pediatric intensive care unit patients with haloperidol was reported by Schieveld and colleagues [70].

On the other hand, atypical antipsychotics such as risperidone, olanzapine, and ziprasidone may be prescribed as alternatives to haloperidol for the treatment of delirium [66,71,72,73]. These drugs act on dopamine receptors, as well as on serotonin, acetylcholine, and norepinephrine neurotransmission [74,75].

All antipsychotics have potentially serious side effects, including extrapyramidal movement disorders, malignant hyperthermia, dysregulated metabolism, laryngeal spasm, constipation, urinary retention, and xerostomia [76]. Although atypical antipsychotics are associated with fewer side effects [77], the lack of well-designed randomized trials on the efficacy of either typical or atypical antipsychotics for the treatment or prevention of delirium in critically ill patients still represents a serious gap [78].


*Q5: What is the best available research evidence on preventing and treating depression and anxiety symptoms in oncopediatric patients?*


While the evidence base in child psychiatry is already small, data on the use of these medications in pediatric oncology are even smaller. Therefore, we concisely review the most up-to-date data about antidepressant efficacy, tolerability, and safety in pediatric patients [79] and suggest their applicability in pediatric oncology, indicating medications from which to choose, and how to start and stop the antidepressant medication.

We briefly consider benefits, risks, the black-box warning, and the significance of off-label prescribing (Discussed below) (Table 2 and Table 3).

Antidepressants are the first line in the treatment of major depressive disorders and anxiety disorders, as well as post-traumatic stress disorders, which are all common in children and adolescents, with a collective prevalence ranging from 15% to 30% [80,81,82].

The preferred and often most efficient treatment modality in these age groups is a combination of psychotherapy and psychopharmacology, depending on the nature of the symptoms [83]. While this holds for the pediatric population in general, children and adolescents treated for cancer often suffer from acute and severe manifestations, especially anxiety-related disorders for which immediate interventions are warranted.

While antidepressants provide substantial benefits in the short and long term, delayed therapeutic onset, as well as intolerance, limited effectiveness, and relapse issues, may limit the efficacy of these drugs [62,63,64]. Although antidepressants induce chemical changes within the brain a few hours after their administration, these do not correspond to a clinical change [65,66,67], which usually takes days and weeks to achieve [66]. In other words, despite the immediate increase in monoamine levels following antidepressant treatment, their positive effects on mood only occur weeks later [68,69]. This delay in efficacy represents a difficult clinical problem to overcome [70]. In fact, not achieving remission from depressive symptoms increases the risk of a more chronic and debilitating course of illness [71,72]. On the other hand, starting early at the first manifestation of severe mood-related problems provides a way to reduce the onset of longer-term problems later on [68,73,74,75].

## 4. Recommendations and Critical Issues

As emerged from our questionnaire, many pediatric oncologists report a lack of training and expressed a need to improve their knowledge and facilitate access to information and resources that allow them to better assist their patients in need of acute help for the severe psychological and psychiatric manifestations that cause unnecessary suffering. After a careful analysis of the literature and guided by the real-world attitudes of pediatric oncologists and healthcare professionals and prescribing practices, we recommend six key points for physicians to keep in mind when prescribing psychotropic drugs for young patients suffering from SABEPs. In addition, we draw pediatric oncologists’ attention to three critical issues to be considered (Figure 1).

### 4.1. Recommendations

The recommendations are intended primarily for pediatric oncologists and other healthcare professionals caring for children and adolescents with cancer.

Target symptoms, not diagnoses. Diagnosis can be difficult in children, especially in the case of comorbidity. Although a working diagnosis is helpful to frame expectations and communicate with young patients and their caregivers, treatment should target key symptoms.Remember the technical aspects of pediatric prescribing. The process of consent is mandatory. The participation of young/adolescent patients and their caregivers in the decision-making process related to their (mental) health and treatment with off-label prescribed drugs needs utmost attention. The Medicine Act of 1968 and European legislation make provisions for physicians to use drugs in an off-label or out-of-license capacity or to use unlicensed medicines. However, individual prescribers need to ensure adequate information to support the efficacy, safety, quality, and intended use of a drug before prescribing it. It is recognized that unlicensed applications are often necessary for pediatric practices.Begin with less, go slow, and monitor efficacy and adverse reactions. In out-patient care, the dosage usually commences lower in mg/kg per day terms than in adults. Evidence suggests that various antidepressants and second-generation antipsychotics may offer a tool to treat severe acute problems but present a variety of safety profiles that should be monitored very closely to detect and manage possible side effects in a timely way. Thus, the dose should be gradually increased as needed and interrupted at a dose that adequately controls symptoms with minimum adverse reactions. Finally, the time necessary to produce a clinical effect is to be considered.Keep in mind that multiple medications are often required for oncopediatric patients. Although monotherapy is ideal, childhood-onset illness can be severe and may require psychosocial interventions in combination with more than one treatment. It is important to distinguish between co-pharmacy, which refers to the use of different medications for different disorders or symptoms, and polypharmacy, which is the use of multiple drugs to manage the same problem. Since children often show multiple co-occurring conditions, co-pharmacy is common.Monitor outcomes in more than one setting. The expression of symptoms may be different across settings (e.g., hospital, home, and school). This is particularly important in the case of symptomatic treatments (e.g., delirium). In this context, current innovative digital technologies, by integrating diverse sources of information, offer a unique opportunity to create predictive models of individual vulnerability [76].(Young) patients and family medication education is essential. The first experiences with treatments are critical for long-term outcomes and adherence to the therapy. Delirium, anxiety, and major depression are the most severe and prevalent complications observed in critically ill children and are the result of their medical condition, its treatment, or both, and cause a temporary severe and acute dysfunction at the cognitive and behavioral level [77,78,79]. Interruption of treatment or subsiding of the disease often puts an end to the episode but is often not a desirable option [80]. Thus, patients and their caregivers should not only be educated regarding the therapies and the possible side effects but should also be encouraged to ask for changes to their treatment regimens if necessary.

### 4.2. Critical Issues

Finally, three critical issues need to be considered by pediatric oncologists when prescribing psychotropic drugs for young patients suffering from SABEPs: prescription of off-label drugs, drug–drug interactions, and the gaps that still need to be filled for the safer use of psychotropic drugs in a pediatric oncology setting.

#### 4.2.1. Off-Label Prescribing

Off-label drug prescribing means prescribing drugs for an indication or using a dosage that has not been approved by regulatory agencies, such as the FDA or EMA. Since these regulatory agencies do not regulate the practice of medicine, the use of off-label drugs has become common [18]. Importantly, the term “off-label” does not imply improper, illegal, or contraindicated use [81]. On the contrary, the lack of labeling for a specific disorder, symptom, age group, or dosage does not automatically mean that the use of the drug is inappropriate for that particular use. Instead, it suggests that the inclusion of a specific drug on the label has not been approved by the EMA and/or FDA [82,83,84,85]. Thus, the responsibilities for the prescription of that therapy rely on the prescriber only.

Previous studies reported that, in the United States, up to 60% of the prescribed drugs are off-label, of which a considerable proportion are prescribed to children and adolescents [81,86,87]. This is mostly due to the lack of research [88].

Clinicians are called to balance beneficence (treat and enhance the quality of life) with nonmaleficence (protect patients from unsafe or ineffective treatments). In general, the therapeutic decision should be driven by the best interest of the patient based on the most convincing available evidence [87,89].

Since pharmaceutical companies are not allowed to advertise their medications for off-label use, doctors should prescribe drugs only for indications approved by the FDA and EFSA labeling systems [63,72].

However, we live in an era of global exchange of medical information, which allows healthcare professionals to weigh the risks and benefits of off-label drug use and provide the best possible care for their patients [90,91,92,93,94,95].

Overall, if there is evidence supporting the use of an off-label drug for a specific indication in a particular patient, informed-consent forms should be obtained/signed, and risks, benefits, and alternatives should be explained and discussed with caregivers and, where possible, patients [81,96]. For example, palliative care—consisting mostly of off-label drugs—has long been recognized as a necessary part of the treatment in the pediatric oncology setting, to assist in managing the high symptom burden of these ‘young’ patients [97].

#### 4.2.2. Drug–Drug Interaction

Given the complexity of the mechanisms of action of drugs used in pediatric oncology [98] and the potential interactions they may have with psychotropic drugs, in this section, we underline some drug interactions to watch out for.

First, it should be remembered that, while treatment with chemotherapeutic drugs follows lengthy treatment regimens, the treatment of SEBAPs is generally short-lasting. Therefore, the risk of pharmacodynamic interactions in acute treatment, although rare, may occur when both drugs (i.e., psychotropic and chemotherapeutic drugs) act on the same pharmacological target. Therefore, the mechanisms of action and pharmacological targets of the drugs need to be considered when they are co-administered.

More likely is the risk of pharmacokinetic interactions if the acutely administered drug is metabolized by the same enzymes as the cancer treatment.

Particular emphasis should be placed on compounds that act as inducers, inhibitors, or substrates of the cytochrome P450 system. Most antineoplastic drugs undergo biotransformation by CYP 450 3A4 [99,100] and, when co-administered with antidepressants that exert inhibitory properties on this CYP isoform, such as fluoxetine, paroxetine, sertraline, and fluvoxamine, may impair the antineoplastic drugs’ efficacy and/or induce higher toxicity. On the other hand, among the antidepressant drugs, escitalopram, citalopram, mirtazapine, venlafaxine, and milnacipran represent alternatives with a higher safety profile for oncologic patients because of their weak CYP 450 inhibitory potential [101].

Consequently, while fluoxetine is the most widely approved SSRI for the treatment of depression by the FDA and EMA, the interactions of fluoxetine with other drugs constitute an important drawback for its use in pediatric oncology patients [102,103,104]. Sertraline, citalopram, and escitalopram are characterized by substantially fewer drug interactions and are therefore preferred [59,105,106].

Citalopram, especially, is secure and effective [107], whereas the use of monoamine oxidase inhibitors should be avoided because of their adverse interactions with other substances [108].

#### 4.2.3. Barriers to Mental Health Services in Pediatric Oncology

To date, there are still some gaps that need to be filled for the safer use of psychotropic drugs in a pediatric oncology setting:

(1) Lack of information related to psychiatric manifestations in the pediatric oncology setting.

(2) Limitations of the acute setting that influence the timely and comprehensive evaluation of emotional-behavioral problems.

(3) Lack of education and training of the medical and mental health staff regarding the identification and management of pediatric psychiatric manifestations.

## 5. Conclusions and Considerations

Overall, a major lack of information exists regarding the use of psychotropic medication for acute severe problems in the pediatric oncology setting. Furthermore, a major difficulty surrounded the appropriate identification of symptoms displayed by children and adolescents with cancer. Beyond symptoms that cluster together, many factors affect the pattern and magnitude of the emotional, cognitive, and behavioral response of the child or adolescent both to disease and treatment, including the age of onset, the duration of treatment, the type of treatment, sex, and genetic background of the child/adolescent, as well as the psychosocial setting of the patient and their family. Furthermore, future studies should consider some characteristics of healthcare professionals, such as age, work experience, sex, and geographic context, as potentially influencing prescribing practices.

Growing evidence stresses the fact that (early) experiences are incorporated into the developing brain. When young children and adolescents are exposed to major adversity, maturing brain circuits can be harmed, and the risk for problems in learning, behavior, and mental health is significantly enhanced [109,110]. Therefore, policy and practice in our centers should focus more on protecting children and adolescents from the neurodevelopmental consequences of the adversities posed by their disease and its treatment. Psychopharmacological treatment of acute and severe problems may help alleviate the damaging effects of significant and repetitive adversity.

In the context of the social, biological, and psychological dimensions of care for pediatric cancer patients, pediatric oncologists, in the absence of developmental psychiatrists, need to be able to distinguish and manage possible acute and severe neuropsychiatric conditions. Especially children and adolescents treated for acute lymphoblastic leukemia and those diagnosed and treated for a brain tumor display a heightened association with significant comorbid psychopathology and exhibit the most extensive behavioral and emotional complications [111].

Here, we describe an approach to managing severe and acute behavioral and emotional problems in pediatric oncology that is useful for physicians considering the complexity of the illness and the psychopathological problems and side effects of the disease and its treatment. Together with pain, comorbid psychiatric disorders represent a significant source of distress, often impairing functioning. The pharmacological treatment of these disorders can feel challenging because patients are typically young and rather sick. However, a full recovery from their problems is often possible even in the most severely ill children and adolescents if we act in a timely way.

Our efforts have the purpose of heightening the collective awareness of this type of devastating concurrent problem and hopefully result in intensifying the endeavors to improve our understanding of the long-term consequences of these problems if left untreated, as well as lead to exploring potentially promising protective treatment regimens. The recommendations of this working group do not represent a formal consensus statement but, as stated before, are the result of several cycles of continuous consultation before reaching an agreement among all members and co-authors.

Finally, further interdisciplinary multi-center research is necessary to investigate the effects of SABEPs on long-term outcomes related to the burden of disease and mental health. Additional efforts should be dedicated to studying the effects of the short-term use of psychotropic drugs while being treated for a tumor, as a possible preventive or protective intervention. Incorporating a framework of standardized screening and developing individual-based interventions, which may include psychotropic drugs when the problem is severe and acute, is an essential investment because it may attenuate long-term problems. In the meantime, our short-term efforts should focus on advocacy and education. Due to the small numbers, children and adolescents with different types of cancers are grouped into broad categories that are not clinically meaningful from a mental health point of view.

Demonstrating a clear causal relationship between emotional and behavioral problems and global or specific neurodevelopmental outcomes for treatments that include multiple drugs exerting cytotoxicity through several different pathways may appear as an almost impossible challenge. However, the treatment of acute behavioral manifestations of toxicity should be considered a critical problem and would benefit from treatment recommendations that integrate different fields of knowledge, such as neurodevelopmental physiology, biochemistry, pediatric pharmacology, and genetics. The brain of children and adolescents suffering from cancer undergoes a series of events that can be defined as a “multiple hit” model, where the type of cancer, its treatment, being diagnosed and hospitalized for it, and exposure to repeated painful procedures all interact with the genetic make-up and resilience or sensitivity to stress of the patients, where the sensitivity to environmental influences determines vulnerable phenotypes [109,112,113].

Finally, the translation of research-based knowledge into clinical care guidelines or recommendations is becoming increasingly important to direct our attention to translating knowledge from research to the development of intervention-based approaches, which should be designed to avoid, prevent, or ameliorate adverse outcomes. These interventions should include social and behavioral, but, when necessary, in acute stages, neuro-pharmacological approaches.

In particular, the results of our questionnaire highlight the need to increase the amount and dissemination of information, education, and training related to the management of acute mental health-related problems presented by children and adolescents suffering from cancer. This, then, may enable the pediatric oncologist to act in a timely and the best possible way, thereby avoiding unnecessary suffering. More importantly, we should evaluate new ways and methods to increase communication and collaboration among pediatric oncologists and mental health specialists, who are called to collaborate to improve survivorship care and the quality of life of young patients.

Given the young age of pediatric cancer patients and survivors, as well as their potential longevity, delayed consequences of therapy are likely to have a greater impact on their lives and families than the acute complications of treatment. The timely provision of interventions to remediate or prevent complications can help monitor neurodevelopment and offer a significant step in reducing any possible risk.

In sum, this comprehensive and updated review on the use of psychotropic drugs for the treatment of SABEPs in children and adolescents treated for cancer underlines the need for close collaboration among pediatric oncologists, child mental health professionals, and other experts. In the presence of risk and uncertainty in the realm of care for pediatric cancer patients, asking the ‘right questions at the right time’ is fundamental.

We think and anticipate that by providing a practical tool, we might assist pediatric oncologists and mental health professionals, as well as pediatric patients and their caregivers, in the decision-making process related to what, when, who, and for whom to prescribe psychotropic drugs in the pediatric oncology setting.

## Figures and Tables

**Figure 1 children-09-01878-f001:**
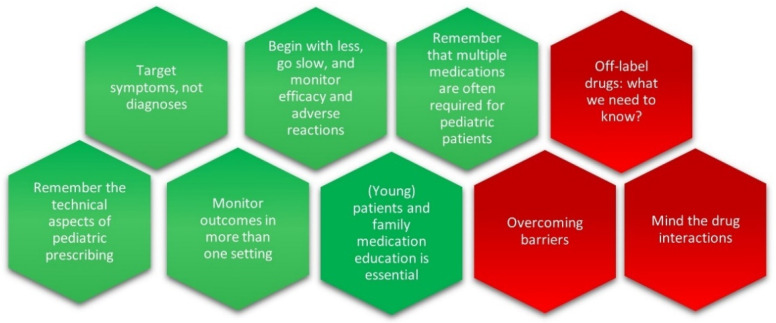
Recommendations are in green and critical issues are in red.

**Table 1 children-09-01878-t001:** Antipsychotic drugs used and recommended in the Italian pediatric haemato-oncology setting.

Antipsychotic Drug	Age	Recommended Dose
Chlorpromazine (approved for ≥ 18) *	1 year and 5 months	3 drops 3 times/die Up to 6 drops, 3 times/die
Promazine (approved for ≥ 18) *	>6 years old	1 drop twice a day
Haloperidol (approved for ≥ 18) *	pre-puberty post-puberty	0.5–8 mg/die 1–16 mg/die
Risperidone (approved for ≥ 18) *	>10 years old	1 mg then in steps 0.75 for 3 days and 0.5 for the following range 0.25–6 mg/kg
Olanzapine Quetiapine Aripiprazole (approved for ≥ 18) *	>6 years old	At low dose 2.5–20 mg/kg

***** Note: the approval of the drugs refers to the EMA.

**Table 2 children-09-01878-t002:** Antidepressant drugs used and recommended in the Italian pediatric haemato-oncology setting.

Antidepressive Drug	Age	Recommended Dose	Side Effects
Fluoxetine Fluvoxamine (approved for ≥ 18) *	>5 years old	10–20 mg/die 50–200 mg/die	Headaches, gastrointestinal side effects, feeling jittery, disinhibited, activated impulsivity, agitation, and suicidality
Citalopram (approved for ≥ 18) *	>6 years old, especially in adolescent patients	10–40 mg/die	Headaches, gastrointestinal side effects, feeling jittery, disinhibited, activated impulsivity, agitation, and suicidality
Escitalopram (approved for ≥ 18) *	pre-puberty post-puberty	5–20 mg/die 1–16 mg/die	Headaches, gastrointestinal side effects, feeling jittery, disinhibited, activated impulsivity, agitation, and suicidality
Sertraline (approved for ≥ 18) *	6–17 years old	12.5–200 mg/die	Headaches, gastrointestinal side effects, feeling jittery, disinhibited, activated impulsivity, agitation, and suicidality
Nortriptyline Desipramine Amitriptyline (approved for ≥ 18) *	>6 years old >12 years old >12 years old	1–3 mg/kg/die 25–100 mg/die 10–200 mg/die	Cardiovascular and anticholinergic side effects. May be lethal in overdose.
Mirtazapine (approved for ≥ 18) *	>6 years old	7.5–45 mg/die	Somnolence, QTc prolongation, weight gain, and agranulocytosis

* Note: the approval of the drugs refers to the EMA.

**Table 3 children-09-01878-t003:** Anxiolytic drugs used and recommended in the Italian pediatric haemato-oncology setting.

Anxiolytic Drug	Age	Recommended Dose	Side Effects
Lorazepam (approved for ≥ 18) *	>10 years old	0.5–2 mg	Sedation, fatigue, dizziness, anticholinergic side effect, and paradoxical activation (in younger children)
Delorazepam (approved for ≥ 18) *	>3 years old	10 drops/day	/
Alprazolam (approved for ≥ 18) *	>14 years old	1 mg × 3 times/die Cp 0.25 2 times/die	/
Clonazepam (approved for ≥ 18) *	>6 years old	0.5 mg/die	Sedation, confusion, and paradoxical activation, particularly in youth with central nervous system dysfunctions

* Note: the approval of the drugs refers to the EMA.

## Data Availability

Not applicable.

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
