# Peer review of "The Use of Psychotropic Medication in Pediatric Oncology for Acute Psychological and Psychiatric Problems: Balancing Risks and Benefits"

_children, 2022, doi:10.3390/children9121878_

Round 1

Reviewer 1 Report

This is a very timely and needed paper as the authors seek to provide a framework to guide the prescribing of psychotropic medications in pediatric oncology patients. It is all too often that prescribing occurs with limited understanding of the utility, and without consideration of more appropriate medications and/or non-pharmacological interventions. The authors (fairly) state this frustration in their comments about an “ideal” vs. “less-than-perfect world”. The flow charts are particularly useful and their immediate utility in hospital settings is clear.

Overall, there are very few criticisms I can make of this paper. I have detailed some small errors identified below as well as some commentary regarding the use of antidepressants. What does appear to be missing though is any guidance around the use of psychostimulants in young patients. As a non-significant proportion of children will premorbidly suffer from disorders of Attention that include behavioral symptoms such as hyperactivity, consideration for guidance around use of these medications is, I believe, warranted. Off-label use of psychostimulants in young patients does also occur in the context of withdrawal and dysphoric mood, as well as in palliation in oncological patients; therefore some mention would be of benefit. Whilst this may occur more in the adult oncology space there is likely some utility to consider these pediatrically.

Page 5, Line 199. It should be “frequently prescribed antipsychotic” rather than “frequently antipsychotic prescribed”.

The authors state pediatric oncology patients (Page 6, lines 243-245) may require immediate interventions. Particularly for anti-depressants, there can be a significant lead time before any efficacy is experienced. Some comment regarding this; if not also a recommendation given this is a guidelines document, would be warranted. The authors do reference this on Page 7, Line 290, however a little more elaboration, particularly as it relates to antidepressants, would be useful given – as the authors state – there is often a very limited understanding of psychotropic medications among pediatric oncologists.

Page 7, Line 260 (Figure 1) there are a few errors. It should be “Begin with less” rather than “Being with less”. There is missing text after “are often required for oncopediat…”. An open bracket is needed before “Young” and “s” is missing for the “is” instead of just “i” in “medication education i essential”.

Page 7, Line 283. It should be “Begin with less” not “Being with less”.

Page 8, Line 300. Missing “it” before “is important”.

In Appendix 1, the last bullet point in the top-right box (Line 484) appears to be in Italian.

In Appendices 1-4, the last bullet of the box (mid-far right) “Monitor on a regular basis” lists “Side effects (see Appendix 2). It is not clear where or what in Appendix 2 the authors are referring to. It seems there may be an additional appendix missing detailing this information and/or management of side effects from psychotropic medication.

Author Response

This is a very timely and needed paper as the authors seek to provide a framework to guide the prescribing of psychotropic medications in pediatric oncology patients. It is all too often that prescribing occurs with a limited understanding of the utility, and without consideration of more appropriate medications and/or non-pharmacological interventions. The authors (fairly) state this frustration in their comments about an “ideal” vs. “less-than-perfect world”. The flow charts are particularly useful and their immediate utility in hospital settings is clear. Overall, there are very few criticisms I can make of this paper. I have detailed some small errors identified below as well as some commentary regarding the use of antidepressants.

We would like to thank Reviewer 1 for her/his comments on our manuscript and for the acute suggestion provided, which allowed us to improve the quality of our paper.

Reviewer 1’s comment: What does appear to be missing though is any guidance around the use of psychostimulants in young patients. As a non-significant proportion of children will premorbid suffer from disorders of Attention that include behavioral symptoms such as hyperactivity, consideration for guidance around the use of these medications is, I believe, warranted.

Authors’ response: We thank Reviewer 1 for her/his pertinent comment. We added a phrase to make pediatric oncologists aware of this important issue, which reads as follows: Also, a small group of patients comes with pre-existing behavioral problems such as hyperactivity attention disorders, for which are treated with psychostimulants and other compounds [34–36]. As for the other psychotropic medications, pediatric oncologists need to pay attention when selecting the appropriate therapy, especially about possible interactions with the cancer treatment proposed, avoiding the combination of these treatments creating SABEPs [37].

Reviewer 1’s comment: Off-label use of psychostimulants in young patients does also occur in the context of withdrawal and dysphoric mood, as well as in palliation in oncological patients; therefore some mention would be of benefit. Whilst this may occur more in the adult oncology space there is likely some utility to consider this pediatrically.

Authors’ response: We thank Reviewer 1 for her/his pertinent comment. We agree and added a specific sentence, which reads as follows: For example, palliative care – consisting mostly of off-label drugs - has long been recognized as a necessary part of the treatment in the pediatric oncology setting, to assist in managing the high symptom burden of these ‘young’ patients [83].

Reviewer 1’s comment: Page 5, Line 199. It should be “frequently prescribed antipsychotic” rather than “frequently antipsychotic prescribed”.

Authors’ response: Reviewer 1 is correct. We have now corrected the sentence as suggested.

Reviewer 1’s comment: The authors state pediatric oncology patients (Page 6, lines 243-245) may require immediate interventions. Particularly for anti-depressants, there can be a significant lead time before any efficacy is experienced. Some comment regarding this; if not also a recommendation given this is a guidelines document, would be warranted. The authors do reference this on Page 7, Line 290, however a little more elaboration, particularly as it relates to antidepressants, would be useful given – as the authors state – there is often a very limited understanding of psychotropic medications among pediatric oncologists.

Authors’ response: Reviewer 1 raised an important point here. Whilst antidepressants provide substantial benefits in the short and long term, intolerance, delayed therapeutic onset, limited effectiveness, and relapse issues represent important factors that may limit the efficacy of these drugs. Although a few hours after their administration, antidepressants induce chemical changes within the brain, these do not correspond to clinical change, which usually takes days and weeks to achieve. In other words, despite the immediate increase in monoamine levels following antidepressant treatment, their positive effects on mood occur only weeks later. This delay in efficacy represents a difficult clinical problem to overcome. In fact, not achieving remission from depressive symptoms increases the risk of a more chronic and debilitating course of illness. On the other hand, starting early at the first manifestation of severe mood-related problems provides a way to reduce the onset of longer-term problems later on. We added this important consideration in the text of the paper.

Reviewer 1’s comment: Page 7, Line 260 (Figure 1) there are a few errors. It should be “Begin with less” rather than “Being with less”. There is missing text after “are often required for oncopediat…”. An open bracket is needed before “Young” and “s” is missing for the “is” instead of just “i” in “medication education I essential”. Page 7, Line 283. It should be “Begin with less” not “Being with less”.

Authors’ response: We thank Reviewer 1 for her/his comments. We have now corrected the figure as suggested.

Reviewer 1’s comment: Page 8, Line 300. Missing “it” before “is important”.

Authors’ response: Reviewer 1 is correct. We have now corrected the sentence as suggested.

Reviewer 1’s comment: In Appendix 1, the last bullet point in the top-right box (Line 484) appears to be in Italian.

Authors’ response: We thank Reviewer 1 for her/his comments. We have now corrected the figure accordingly.

Reviewer 1’s comment: In Appendices 1-4, the last bullet of the box (mid-far right) “Monitor regularly” lists “Side effects (see Appendix 2). It is not clear where or what in Appendix 2 the authors are referring to. It seems there may be an additional appendix missing detailing this information and/or the management of side effects from psychotropic medication.

Authors’ response: We thank Reviewer 1 for pointing this out. We have now edited Appendix 2, as required.

Reviewer 2 Report

The manuscript deals with the search for evidence on the use of psychotropic medications in pediatric oncology, due, on the one hand, to the frequency with which children and adolescents with cancer suffer mental problems and, on the other hand, to the lack of consensus regarding pharmacological treatment of these problems. This issue has evident social and clinical interest, and I think the document fits Children journal aims.

For that, a qualitative study was developed, using a Delphi procedure. And the use of this technique represents my concerns about the manuscript, especially related with the absence of information.

-          Who were the experts? Authors mention “…Experts included specialists in pediatric oncology, pediatric and adult psycho-oncology, neurology, pediatric neuroscience, psychiatry, (neu-ro)psychology, and neuropharmacology”. But, in suplementary material say “A total of 28 pediatric oncologists and 12 child mental health specialists responded to the survey”, but only 40% completed the survey. Can authors provide who really participated as experts, and their specific speciality? Are there some bias in specialities' representation? There are some reasons why the 60% refused to participate? In this type of methodology informant quality is essential.

-          I am understanding a semi-closed adapted questionnarie at first stage/phase was administered. Usually, general questions are used as a preliminary round to detect general needs / issues. How many stages were used?

-          How information was extracted in each stages? There was used a specific software?

-          Can authors provides the final questionnarie used?

-          A final concern: Authors mention “A review of the literature was performed to provide an overview of the current state of the art including the presence of barriers and problems”. Two questions: (i) I am not sure if the results and recommendations only came from experts’ opinions, or if this literature review are part of those final recomendations (together the experts' advices); (ii) more methodological information is needed about this feasibily (bias control) of the review, especially if its data are affecting recommendations.

-          I think manuscript provides relevant information, and can be accepted, but I feel methodological issues need to be cleared for result verificability, far from arbitrary decisions.

Author Response

Reviewer 2’s comment: The manuscript deals with the search for evidence on the use of psychotropic medications in pediatric oncology, due, on the one hand, to the frequency with which children and adolescents with cancer suffer mental problems and, on the other hand, to the lack of consensus regarding pharmacological treatment of these problems. This issue has an evident social and clinical interest, and I think the document fits the Children’s journal’s aims. For that, a qualitative study was developed, using a Delphi procedure. And the use of this technique represents my concerns about the manuscript, especially related to the absence of information.

Authors’ response: We thank Reviewer #2 for his/her insightful comments and remarkable suggestion on our manuscript, which allowed us to improve the quality of our paper.

Reviewer 2’s comment: Who were the experts? The authors mention “…Experts included specialists in pediatric oncology, pediatric and adult psycho-oncology, neurology, pediatric neuroscience, psychiatry, (neuro)psychology, and neuropharmacology”. But, supplementary material says “A total of 28 pediatric oncologists and 12 child mental health specialists responded to the survey”, but only 40% completed the survey.

Authors’ response: We thank the Reviewer for her/his comments. The experts in the workgroup included experts in pediatric oncology (Barisone, Caprino, Cellini, Jancovic, and Pancaldi), psycho-oncology (Bertolotti, Clerici, Di Giuseppe, Favara- Scacco, Scarponi, and Torta) neurology (Torta), pediatric neuroscience (Blom), psychiatry (Pani), (neuro)psychology (Poggi), and neuropharmacology (Colliva, Pani, Rivi, Tascedda, and Torta), who authored this manuscript.

The questionnaires were sent out to Italian pediatric oncology centers and a total of 28 pediatric oncologists and 12 child mental health specialists (including psychologists and psychiatrists) responded to the survey. We apologize that we did not fully write the percentages of responders: participants could answer online or in paper format; 40% completed the survey online and 60% responded in paper format. Following the Reviewer’s 2 suggestions, we have now edited the Supplementary material sections.

Reviewer 2’s comment: Can authors provide who participated as experts and their specific specialty? Is there some bias in specialties’ representation? There are some reasons why 60% refused to participate. In this type of methodology, informant quality is essential.

Authors’ response: We privileged the larger participation of pediatric oncologists and psycho-oncologists, but we completed the workgroup with pertinent other fields of expertise. Having the ex-director of AIFA (Italian Medicines Agency) among the experts, providing an important safeguard regarding regulatory aspects of prescribing antipsychotic drugs to minors.

Reviewer 2’s comment: I am understanding a semi-closed adapted questionnaire at the first stage/phase was administered. Usually, general questions are used as a preliminary round to detect general needs/issues. How many stages were used?

Authors’ response: We thank Reviewer 2 for pointing this out. The study started because of the many problems encountered in a clinical setting and reported in a non-formal way. As a consequence, we created an intersociety group (the Italian Society of pediatric hematology and oncology and the Italian Society of Psycho-oncology) and reached with experts. This group met in the first round of consultation and decided that the questionnaire was needed to better understand the entity of the problem. Based on the results of the questionnaire, the second round of consultation was performed and clinical questions emerged.  Thus, the experts in the workgroup were asked to answer the clinical questions, providing detailed and updated information. A final round of consultation was then performed to develop the recommendations reported in this study and ask attention to possible barriers and problems related to this complex topic.

We have now reported the specific design of our consultations in the main text.

Reviewer 2’s comment: How information was extracted in each stage? There was used a specific software?

Authors’ response: Information was extracted by three experts (Blom, Pancaldi, and Rivi) from the recordings of the workgroup’s meeting. No specific software was used. Data extracted from the meetings were shared among experts for completeness.

Reviewer 2’s comment: Can the authors provide the final questionnaire used?

Authors’ response: The questionnaire attached in the Supplementary material is the sole questionnaire used. The second round of consultation resulted in clinical questions to guide pediatric oncologists in their choice of treatment. As such, the evidence sustaining these questions must be periodically reviewed and updated to provide new and pertinent evidence.  

Reviewer 2’s comment: A final concern: Authors mention “A review of the literature was performed to provide an overview of the current state of the art including the presence of barriers and problems”.  Two questions: (1) I am not sure if the results and recommendations only come from experts’ opinions, or if this literature review is part of those final recommendations (together with the experts' advice); (2) more methodological information is needed about the feasibility (bias control) of the review, especially if its data are affecting recommendations.

Authors’ response: The results and recommendations reported integrated experts’ opinions with the most recent and updated data concerning treatment regimens and regulations. Bias was avoided by separating the review of the literature from workgroup discussions. Thus, the state of the art was not driven by any expert personal opinion.

Reviewer 2’s comment: I think the manuscript provides relevant information and can be accepted, but I feel methodological issues need to be cleared for result verifiability, far from arbitrary decisions.

Authors’ response: Again, we thank Reviewer #2 for his/her insightful comments and remarkable suggestion on our manuscript, which allowed us to improve the quality of our paper.

Round 2

Reviewer 2 Report

I consider the authors have replied to the suggestions made and have clarified my main concerns.